# Nickel/biimidazole-catalyzed electrochemical enantioselective reductive cross-coupling of aryl aziridines with aryl iodides

Yun-Zhao Wang[1,4], Zhen-Hua Wang[1,4], Inbal L. Eshel [ORCID][2,4], Bing Sun[1], Dong Liu[1], Yu-Cheng Gu [ORCID][3], Anat Milo [ORCID][2] ✉ & Tian-Sheng Mei [ORCID][1] ✉

Here, we report an asymmetric electrochemical organonickel-catalyzed reductive cross-coupling of aryl aziridines with aryl iodides in an undivided cell, affording β-phenethylamines in good to excellent enantioselectivity with broad functional group tolerance. The combination of cyclic voltammetry analysis of the catalyst reduction potential as well as an electrode potential study provides a convenient route for reaction optimization. Overall, the high efficiency of this method is credited to the electroreduction-mediated turn-over of the nickel catalyst instead of a metal reductant-mediated turnover. Mechanistic studies suggest a radical pathway is involved in the ring opening of aziridines. The statistical analysis serves to compare the different design requirements for photochemically and electrochemically mediated reactions under this type of mechanistic manifold.

With the renaissance of organic electrolysis[1–8], asymmetric electrocatalysis involving anodic oxidation has attracted extensive interest in recent years[9–28]. However, less attention has been paid to asymmetric electrochemical reduction reactions[29–32]. In 1997, Durandetti developed the first example of asymmetric electrochemical cross-coupling using chiral auxiliaries[29]. In 2019, a Ni/Box catalytic system for the cross-coupling of alkenyl and benzyl halides without super-stoichiometric metal powder reductants was reported by Reisman and coworkers (Fig. 1a)[30]. Our group described a Ni/Pyrox catalyzed electrochemical reductive homocoupling of aryl bromides, allowing the readily scalable synthesis of enantioenriched axially chiral BINOL derivatives[31]. Very recently, Baran and coworkers demonstrated an electrochemical Nozaki-Hiyama-Kishi coupling reaction with a chiral sulfonamide ligand, providing access to chiral alcohols (Fig. 1b)[32]. The scarcity of asymmetric electrochemical reduction reactions can probably be ascribed to the intolerance of chiral catalysts to reductive electrochemical conditions.

Inspired by the pioneering work of Hillhouse[33], Wolfe[34], Alper, and others[35–53], we wished to develop an electrochemical reductive approach for the cross-coupling of aziridines. Specifically, they showed that transition metal-catalyzed ring expansions[35–38] and ring opening[39–53] of aziridines could enable the mild synthesis of β-lactams, β-phenethylamine, and β-amino acids. Aryl aziridines are versatile synthetic intermediates for the preparation of optically pure 2,2-diarylethylamine scaffolds, which are prevalent in bioactive molecules and pharmaceuticals, including dopamine receptor agonists[54,55]. Although highly regioselective transformations of aziridines have been achieved, enantioselective transformations are challenging and less explored, probably due to the propensity for stereospecificity in transition metal-catalyzed nucleophilic ring openings[46]. Until now, only one example of asymmetric reduction of racemic aryl aziridines and aryl iodides has been realized, namely by Doyle and coworkers, using a nickel/BiOx system in the presence of Mn as the reductant[51]. However, 3–4 days are required to complete the transformation due to the slow turnover-limiting reduction of the Ni catalyst by Zn[56,57]. This important precedent indicated that accelerating the reduction of the catalyst is key to improving the efficiency of this reaction[58–66]. The efficacy of electrochemistry in the reduction of transition metal

---

[1]State Key Laboratory of Organometallic Chemistry, Shanghai Institute of Organic Chemistry, University of Chinese Academy of Sciences, CAS, Shanghai, China. [2]Department of Chemistry, Ben-Gurion University of the Negev, Beer-Sheva 841051, Israel. [3]Syngenta, Jealott's Hill International Research Centre, Berkshire RE42 6EY, UK. [4]These authors contributed equally: Yun-Zhao Wang, Zhen-Hua Wang, Inbal L. Eshel. ✉e-mail: anatmilo@bgu.ac.il; mei7900@sioc.ac.cn

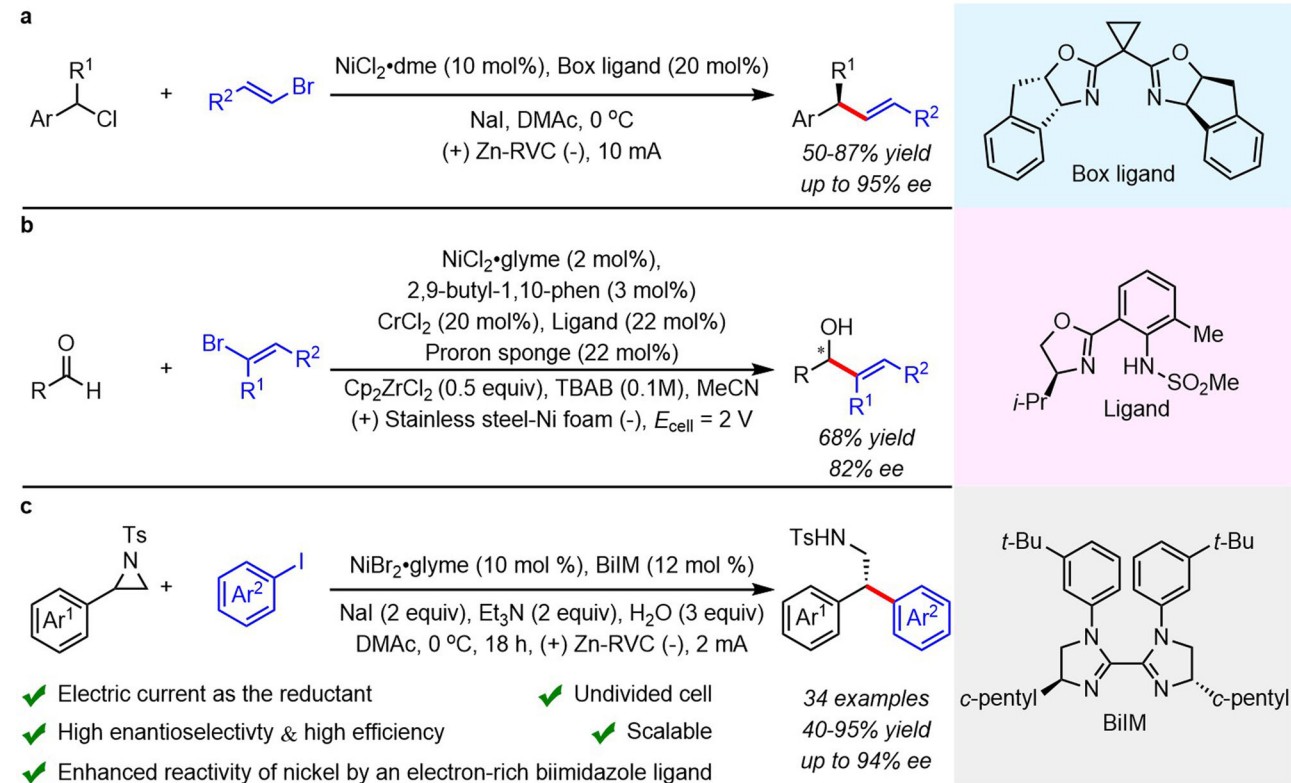

**Fig. 1 | Enantioselective electrochemical reductive cross-couplings.**
**a** Pioneering work on enantioselective electrochemical reductive cross-couplings by Reisman. **b** Enantioselective electrochemical NHK coupling by Reisman, Blackmond, and Baran. **c** This work: Enantioselective electrochemical reductive cross-coupling of aryl aziridines to aryl iodides.

catalysts[67–74], as part of our program[31,75–78], encouraged us to develop transition metal-catalyzed electrochemical reactions.

Herein, we report the first example of enantioselective Ni/BiIM-catalyzed electrochemical reductive cross-coupling of aryl aziridines to aryl iodides (Fig. 1c). Specifically, the pre-activation procedure and requirement of an activating agent for the metal reductant in traditional reduction cross-couplings were avoided by using electrons to turn over the catalyst. Moreover, the limited compatibility of chiral catalysts in an electrochemical cell was resolved by identifying a ligand scaffold that is both selective and stable.

## Results
### Optimization studies
Initially, different chiral ligands were examined for the coupling of 2-phenyl-1-tosylaziridine (**1a**) to 1-iodo-4-methoxybenzene (**2a**) (Fig. 2A). The use of **L1**, which was previously employed by Doyle and coworkers, afforded **3a** in less than 5% yield with 81% enantioselectivity (entry 1)[51]. Other chiral ligands such as pyridine bisoxazoline, pyridine oxazoline, and bisoxazoline all gave low yield and enantioselectivity of **3a** (entries 2–4). To our delight, BiIM ligands **L5** and **L6** improved the yield of **3a** to 39 and 70%, respectively (entries 5 and 6)[53,79–83]. The desired 2,2-diarylethylamine **3a** could be obtained in 97% yield and 88% ee using **L7** as the ligand (entry 7), while **L8** resulted in a diminished yield (entry 8). In addition, when the reaction was conducted at 0 °C with an electric current of 2 mA for 18 h, **3a** was isolated in 95% yield with 90% ee (entry 9). At this point, the cathodic potential was measured to be −2.20 V vs. Fc/Fc⁺ (Fig. 2B, black line). Using Pt as the anode instead of Zn was also tested (entry 10), and the cathodic potential was too positive to reduce the catalyst (Fig. 2C). A low yield of **3a** was observed by replacing Zn with reticulated vitreous carbon (RVC, entry 11), which offers a sufficiently high but unstable anode potential (Supplementary Fig. 10). Evaluation of other anode materials

in this transformation also revealed that the cathode potential could be tuned by different anode materials (Supplementary Fig. 7–9).

The improved performance of BiIM ligands compared to BiOx ligands in this system is strikingly different from the trend observed for the system developed by Doyle and coworkers[51] using the Mn reductant. We hypothesized that the stronger binding of electron-rich BiIM ligands was required for obtaining stable electrochemically active nickel species. Namely, the color in the solution of NiBr₂•glyme and **L7** in DMAc changed from light green to reddish-brown within 30 min, whereas the mixture with **L1** remained colorless (Supplementary Fig. 4). Furthermore, the electron-rich nature of **L7** and the stability of the complexes formed with such ligands was also supported by the facile ligand exchange of **L1** ligated nickel complex to **L7** ligated nickel complex (Fig. 2D, and Supplementary Figs. 5 and 6). To support this claim, we sought to identify a molecular feature that could reveal a correlation with the yield observed with different ligands. We were specifically intrigued by the fact that all BiOx ligands led to <5% yield, whereas only one BiIM led to such a low value. We found that the electronic character of the carbon bridge, which likely represents the push-pull balance of the ligand, correlated well with the yield (Fig. 2E and Supplementary Fig. 19).

### Substrate scope
With the optimized reaction conditions in hand, we investigated the generality and limitations of this electrochemical reductive cross-coupling reaction. As shown in Fig. 3, *para*-(**2a–2k**) and *meta*-substituted (**2l–2u**) aryl iodides with a variety of functional groups such as ether (**2a, 2f, 2s**), alkyl (**2b–2d, 2l**), ester (**2e, 2n, 2o, 2t**), aryl (**2g, 2m**), trifluoromethyl (**2h, 2p**), halogen (**2i, 2j, 2q, 2r**), and amino (**2k, 2u**) groups were well tolerated under the electrochemical reductive cross-coupling conditions, affording the corresponding β, β-diarylethylamines in good yields and high enantioselectivities. Multi-

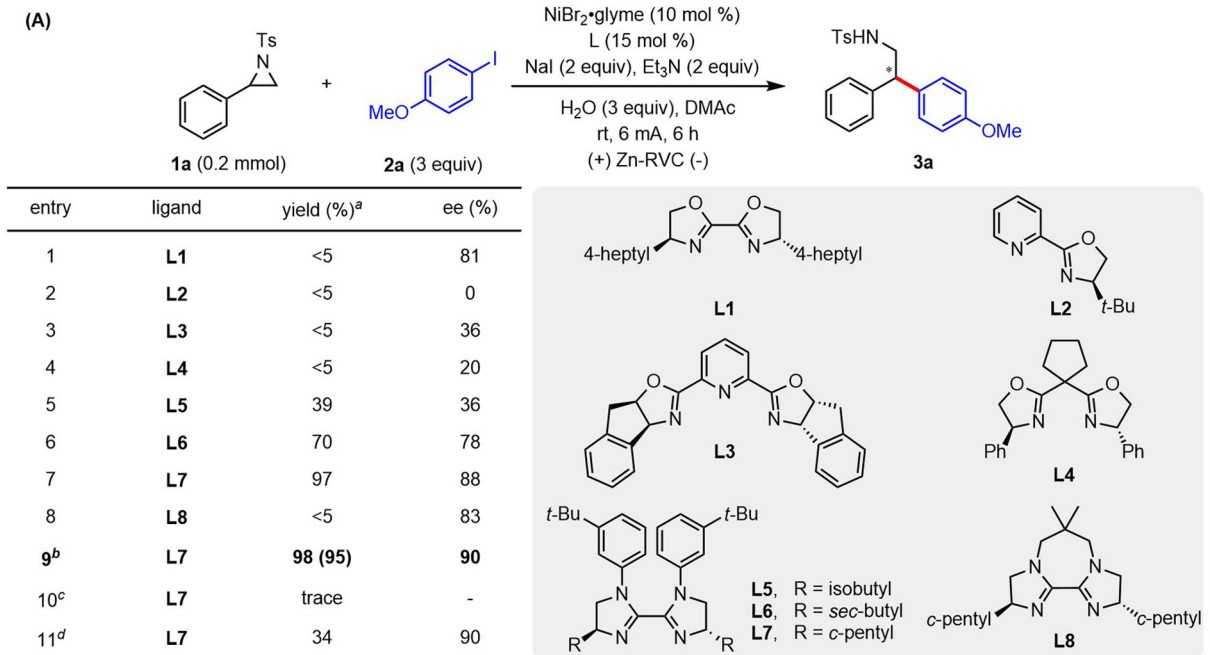

| entry | ligand | yield (%)$^a$ | ee (%) |
|-------|--------|---------------|--------|
| 1 | L1 | <5 | 81 |
| 2 | L2 | <5 | 0 |
| 3 | L3 | <5 | 36 |
| 4 | L4 | <5 | 20 |
| 5 | L5 | 39 | 36 |
| 6 | L6 | 70 | 78 |
| 7 | L7 | 97 | 88 |
| 8 | L8 | <5 | 83 |
| 9$^b$ | L7 | 98 (95) | 90 |
| 10$^c$ | L7 | trace | - |
| 11$^d$ | L7 | 34 | 90 |

$^a$Yields were determined by $^1$H NMR using $CH_2Br_2$ as an internal standard. Isolated yield was shown in parentheses. $^b$2 mA, 0 °C, 18 h. $^c$Platinum (1.0x1.0 cm$^2$) as the anode. $^d$RVC (1.0x1.5x0.3 cm$^3$) as the anode.

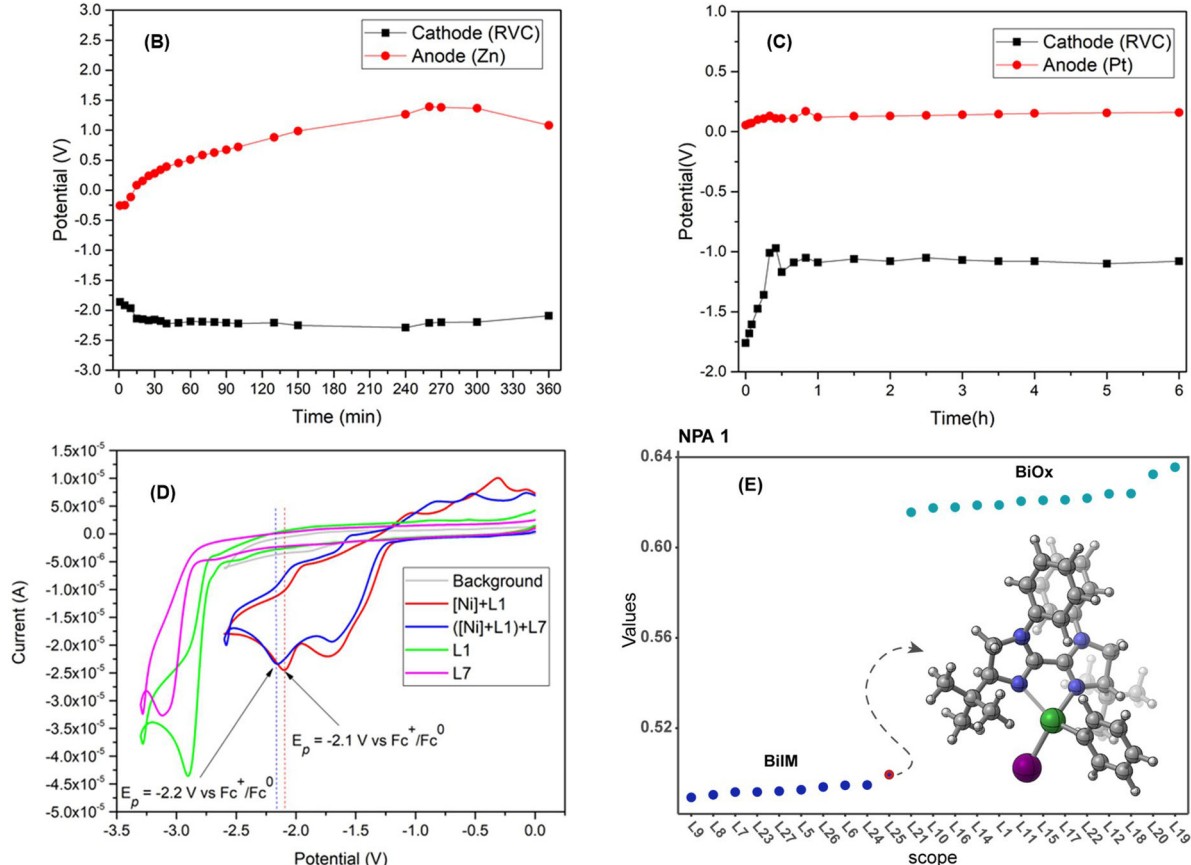

**Fig. 2 | Reaction optimization. A** Ligand evaluation. **B**, **C** Voltaic profiles of the Zn/Pt anode (red) and RVC cathode (black). **D** Cyclic voltammograms were recorded on a glassy carbon WE and Pt CE. DMAc containing 0.1 M $n$-Bu$_4$NPF$_6$ and 5 mM [Ni] and 5 mM **L**$_n$. Potentials were calibrated with Fc as an internal standard. **E** NPA 1, the partial charge of the carbon atom bridge, represents a threshold parameter able to distinguish between the BiIM and BiOx ligands. The y axis represents the values of NPA 1 and the x axis is the scope of ligands tested.

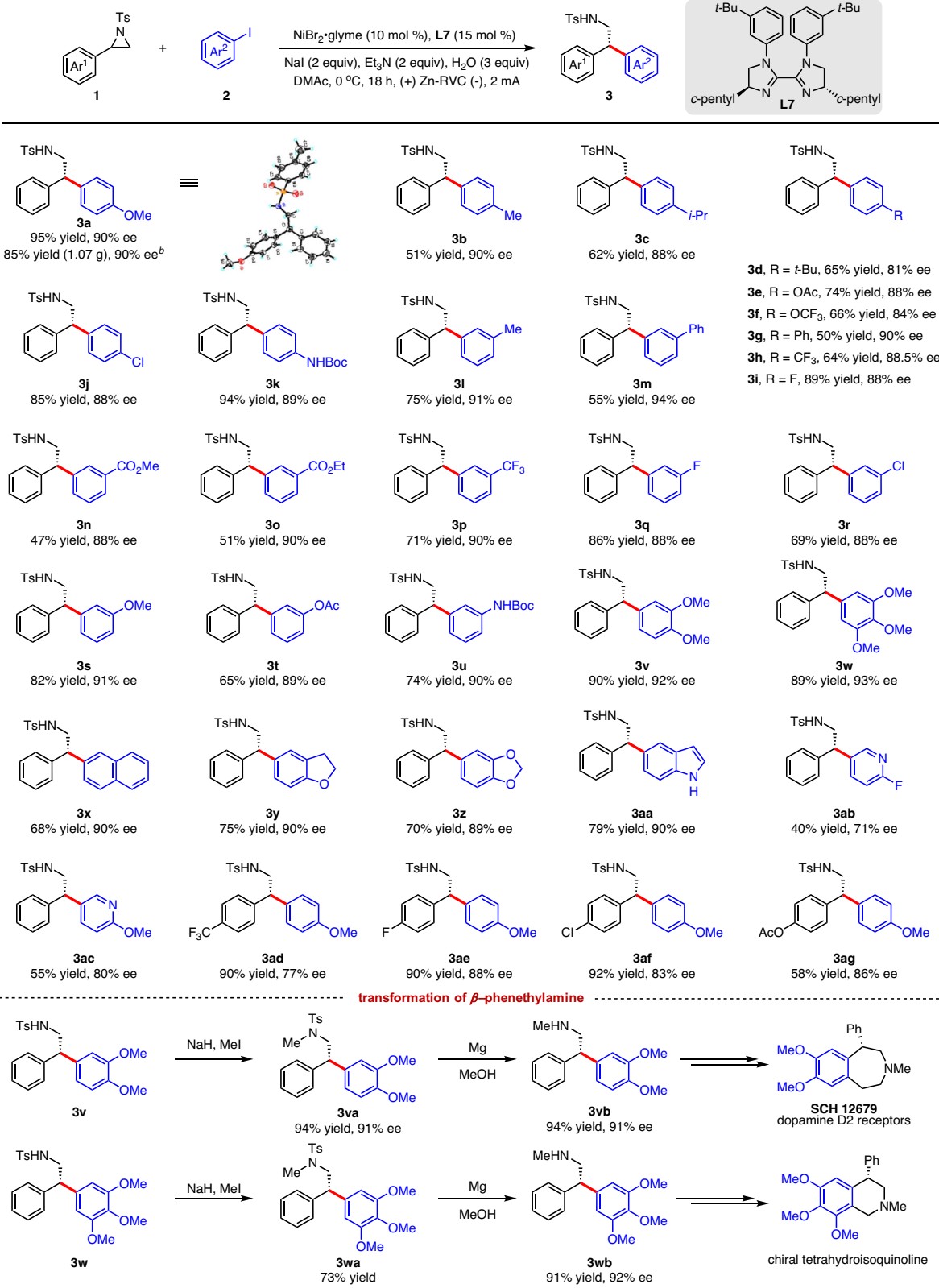

**Fig. 3 | Substrate scope.** [a]All yields refer to isolated product. [b]3.4 mmol scale.

substituted iodobenzenes were converted to the corresponding products (**3v** and **3w**) in excellent yields and enantioselectivities, which could be further derivatized to other biologically active intermediates[84,85]. Hindered 2-naphthoiodobenzene was also effectively coupled in this reaction to give **3x** in 68% yield with 90% enantioselectivity. Heterocyclic iodoarenes were also suitable

coupling partners, affording **3y**−**3ac** in good yields, albeit with generally slight reductions in enantioselectivities. Alkyl iodides such as (2-Iodoethyl)benzene and other alkyl halides were examined under the electrochemical conditions, and no desired products were detected (Supplementary Table 13). Aryl aziridines bearing various functional groups on the arene were also well-suited for this electrochemical

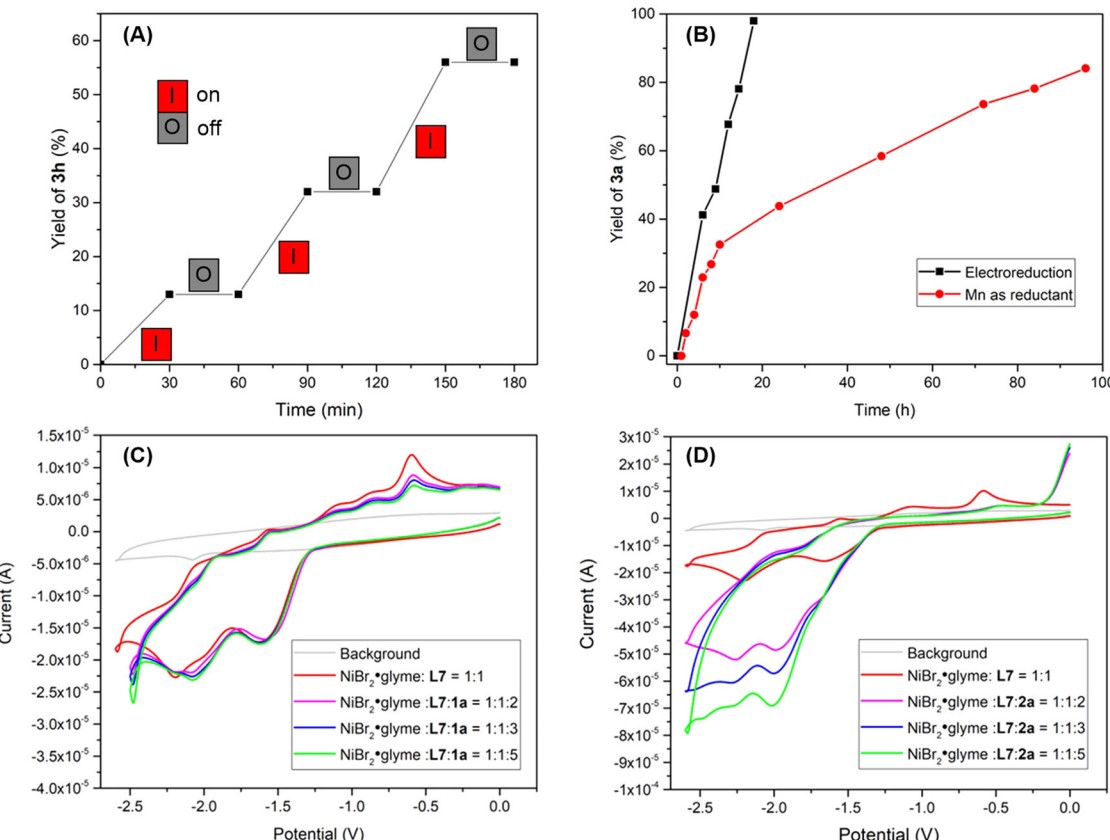

**Fig. 4 | Mechanistic experiments. A** On/off experiment. **B** Time course experiment. **C** CV analysis on the interaction of **1a** with the catalyst. **D** CV analysis on the interaction of **2a** with the catalyst.

reductive cross-coupling reaction, affording **3ad**–**3ag** in good yields and ee values. Notably, **3a** could be synthesized in a gram-scale reaction with 85% yield and 90% enantioselectivity. Heteromatic, 1,2-disubstituted, poly-substituted, and alkyl aziridines were not efficient in this transformation, 2-(naphthalen-2-yl)–1-tosylaziridine and 2-benzyl-1-tosylaziridine were covert to the corresponding ring-opening product in 31 and 28% yield, respectively (Supplementary Table 11 and 12). The absolute configuration of this ring-opening product **3a** was determined by single-crystal X-ray diffraction.

## Discussion

On/off experiments illustrated the crucial role of electricity and show the chemo-reduction of nickel catalyst by Zn does not occur in the background (Fig. 4A). The reaction proceeded more efficiently under electrochemical conditions than Mn powder activated by TMSCl, which further demonstrates the relative efficacy of electroreduction for turning over the nickel catalyst (Fig. 4B). To gain insights into the reaction mechanism, we conducted another series of cyclic voltammetric (CV) analyses. The mixture of NiBr$_2$•glyme and **L7** in a ratio of 1:1 exhibits two quasi-reversible reductive peaks at −1.60 V and −2.20 V vs. Fc/Fc$^+$ in dimethylacetamide, which may be attributed to the reductive potential of Ni$^{II}$/Ni$^I$ and Ni$^I$/Ni$^0$, respectively (Fig. 4C, red line). No significant increase in the reduction peak was observed by the addition of **1a** (Ep = −2.71 V) into the mixture of NiBr$_2$•glyme and **L7** (see Supplementary Fig. 2). In contrast, the reduction peaks of Ni$^{II}$/Ni$^I$ and Ni$^I$/Ni$^0$ were dramatically increased by the addition of **2a** (Ep = −2.69 V, see Supplementary Fig. 2) to the mixture, along with the loss of the oxidation peak (Fig. 4D). These results demonstrate that the catalytic system of NiBr$_2$•glyme and **L7** can oxidatively add to **2a** but not **1a**[51]. Tuning the potential (Fig. 5A): at −1.0 V (Ni$^{II}$), **3a** was not detected; at −1.8 V (Ni$^I$), **3a** was detected in 10% yield; at −2.2 V (Ni$^0$), **3a** was formed

in 68% yield. These constant potential electrolysis experiments illustrate that the oxidative addition of aryl iodides is predominantly caused by Ni(0) species.

Since the oxidative addition of **1a** to nickel species[33] was not observed by CV analysis (Fig. 4C)[51], we carried out the reaction in the absence of the catalyst: no desired product **3a** was detected, whereas 2-vinylnaphthalene and *p*-toluenesulfonamide (TsNH$_2$) were formed with 70 and 68% yield, respectively (Fig. 5B). We presumed that the nucleophilic attack of iodine anion on aziridine leads to its ring opening and the proton formed via oxidation of Et$_3$N[86,87] could act as Brønsted acid to promote the ring-opening process. Treatment of **1a** with Et$_3$N•HI or HI, 2-Iodo-2-phenylmethyl-4-methylbenzenesulfonamide (**4a**) was formed in 81 and 87% yield, respectively (Fig. 5C). Upon subjecting **4a** to our reaction conditions in the absence of the catalyst TsNH$_2$ was formed in 58% yield (Fig. 5D). In the presence of the catalyst, **4a** was readily converted to product **3a** in 67% yield with 90% enantioselectivity (Fig. 5D), which indicates that **4a** is a potential intermediate in this process. To clarify the mechanism of this reaction, we performed the coupling in the presence of radical scavengers (Fig. 5E, entries 1 and 2). Consequently, no reaction occurred in the presence of 2 equivalents of 2,2,6,6-tetramethylpiperidine-1-oxyl (TEMPO) or bis(pinacolato)diboron (B$_2$pin$_2$). Furthermore, both racemic and enantioenriched aziridines afford the enantioenriched product **3a** with equal selectivity (Fig. 5E, entries 3–5). The stereoconvergence of this transformation also supports a radical-based mechanism.

Based on these studies and previous reports[51], a plausible mechanism is presented in Fig. 5F. A [Ni$^0$] species (**B**) may be formed via the cathodic reduction of a [Ni$^{II}$] precatalyst. Oxidative addition of aryl iodide to **B** generates Ar[Ni$^{II}$] species **C**. Concomitantly, nucleophilic iodide ring-opening of **1a** could afford **E**, which can undergo single electron transfer (SET) to create radical **F**. The way of nickel(I)

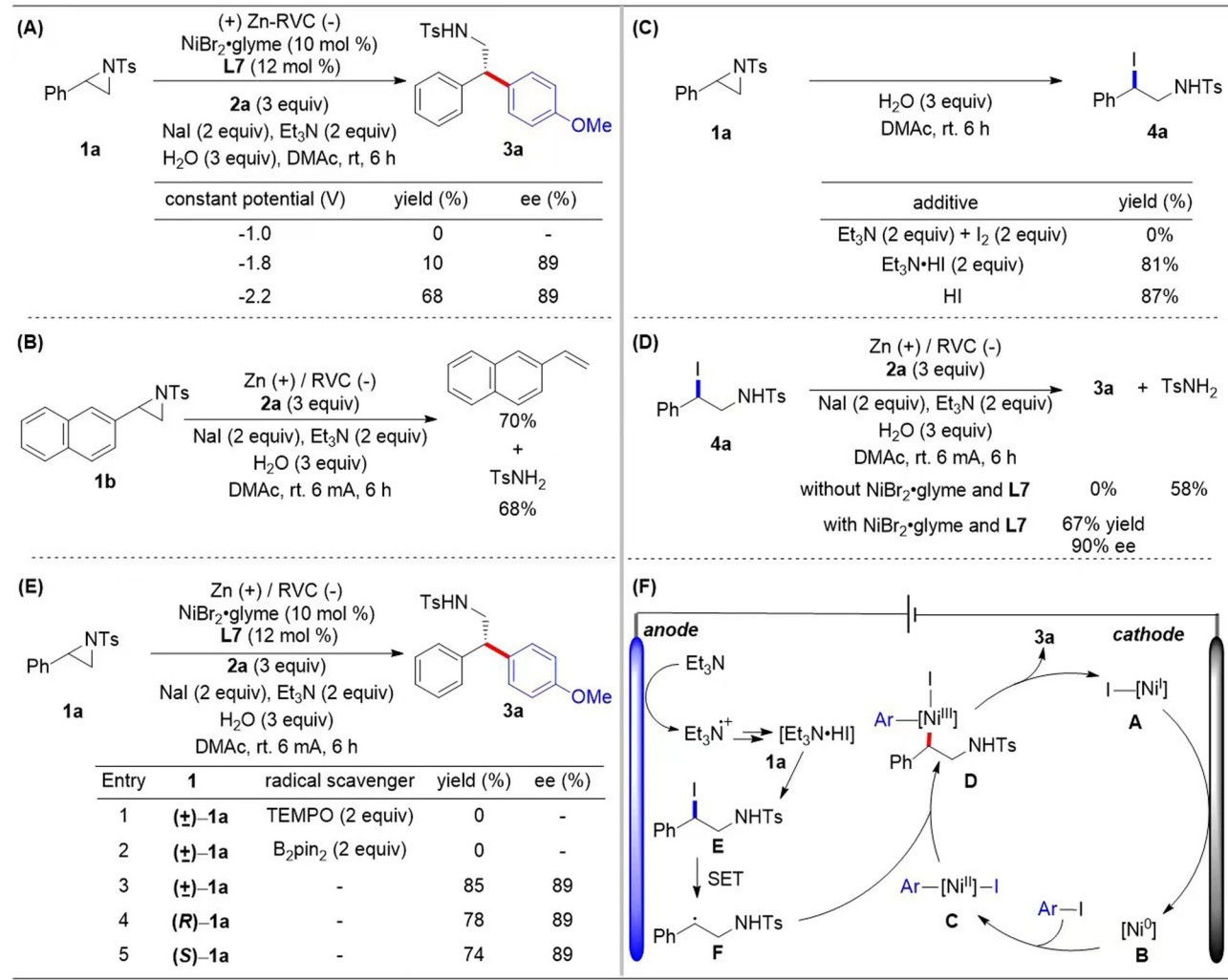

**Fig. 5 | Mechanistic studies. A** Constant potential electrolysis. **B** Control experiment. **C** Ring-opening experiments of aziridine. **D** Transformation of the proposed intermediate. **E** Radical scavenging experiments and stereochemical studies. **F** Proposed mechanism.

activates **E** by single electron transfer can not be ruled out at this stage. This radical intermediate could be trapped by **C** to give [Ni^III] intermediate **D**. Reductive elimination could generate the cross-coupled product **3a** and [Ni^I].

Given this mechanistic proposal, we sought to better understand the role of the ligand structure in the induction of selectivity. We were particularly intrigued by the difference in ligand requirements between this reaction and a mechanistically related photocatalytic reaction developed by the Doyle group (*vide infra*)[53]. To this end, enantioselectivity values for all tested ligands (Supplementary Fig. 15, a total of **21** ligands) were collected to identify a multivariable model based on the structural features of the ligand. According to the proposed mechanism, intermediate **C** (Fig. 5F) was selected for the extraction of structural features across the set of ligands because it directly precedes the enantioselectivity determining step. The computed Ni complex was taken to be square planar, and for simplicity, the aryl group was set as unsubstituted phenyl. The model with the strongest statistical fit that we identified is presented in Fig. 6B. The parameters in this model are both related to the substituents, namely, the distances between the carbons connecting the ligand core to its substituents. We consider this to be a stereoelectronic parameter because it reflects the steric nature of the substituent and the ability of the ligands' core to push or pull electron density from the substituents.

Recently, the Doyle group reported the use of a photocatalyst coupled with a Ni catalyst for the reductive cross-coupling between styrene oxides and aryl iodides[53]. Although they propose a similar mechanism to the one proposed herein (Fig. 5F), the features affecting enantioselectivity in both cases are distinct. The Doyle group identified a model that is mainly influenced by electronic features, which stands in contrast to the more steric nature of the model identified for our electrocatalytic reaction (Fig. 6B, C). To verify that the use of a different parameter set in our work is not at the root of this discrepancy, we set out to identify a model for the enantioselectivity outcomes obtained by the Doyle group using our parameter set. The model with the strongest statistical fit identified for this set is presented in Fig. 6C. As anticipated and in accordance with the model identified by the Doyle group, the parameters featured in this model are mainly electronic in nature. Moreover, the parameters are focalized on the metal center. Specifically, the NPA charge of the metal and its distance from the iodine and nitrogen all play a similar role in the prediction of enantioselectivity.

Indeed, the parameters that describe the photocatalytic reaction reveal a much more electronic character. We assume that the enantioselectivity divergence using the same ligands in these two reactions (see ligands highlighted in light blue, Fig. 6B, C) stems from these distinct features. Thus, it seems that the selection of optimal ligands in both cases should be guided by different considerations. Moreover, the range of enantioselectivity values is different, revealing a much broader spread for the electrochemical reaction, which may allude to a strong dependence on catalyst structure. Indeed, once the optimal ligand was identified, a broad scope was attained, and the

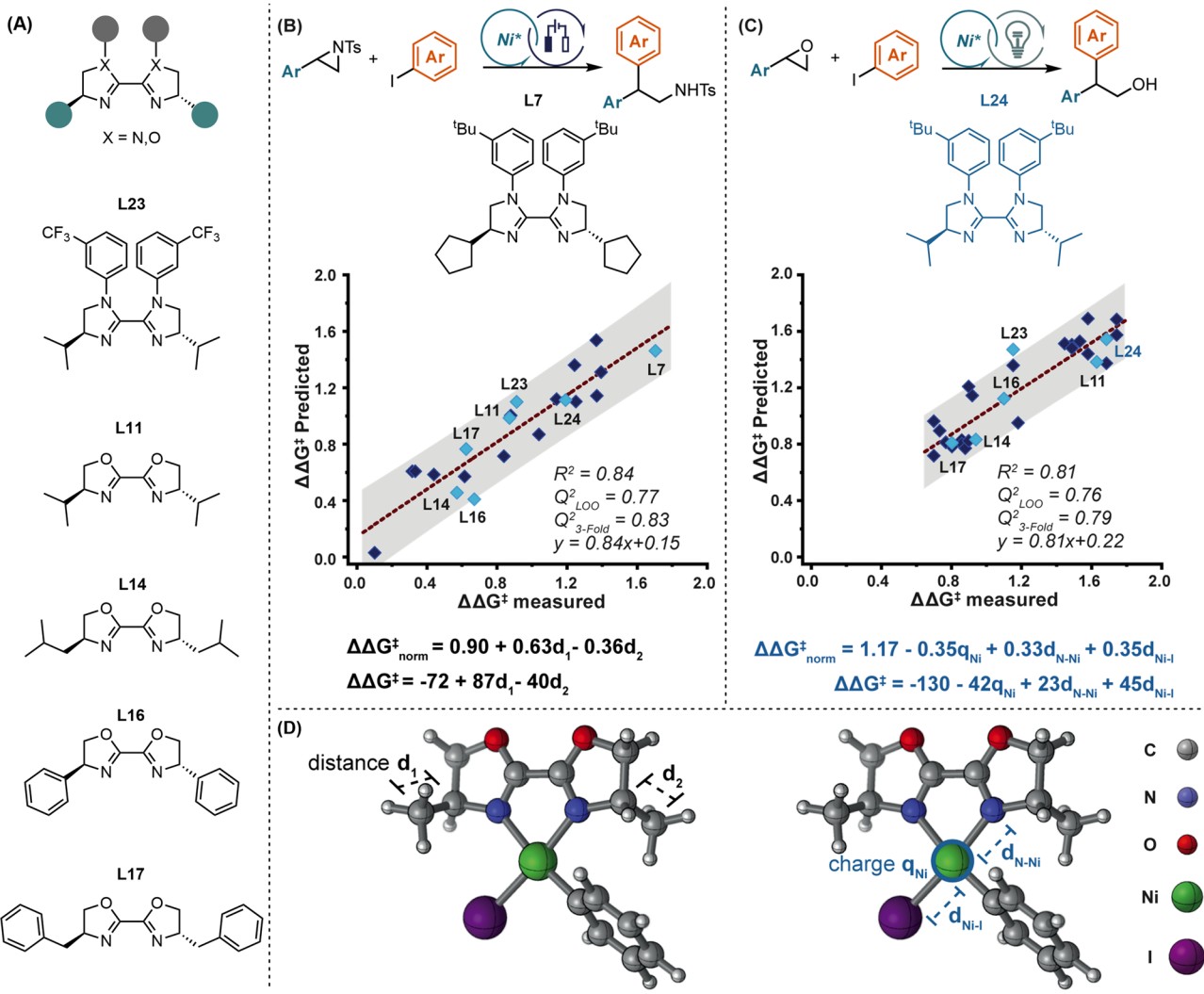

**Fig. 6 | Computational and statistical analysis. A** Examples of ligands used in both studies. **B** Model for the electrochemical reaction. **C** Model for the photochemical reaction. **D** 3D representation of the ligand.

enantioselectivity remained fairly high throughout the whole set. Ultimately, this comparison highlights differences between these seemingly innocuous methods for SET and the utility of having both in the synthetic toolbox.

In summary, we have presented the first example of an enantioselective nickel-catalyzed electrochemical reductive cross-coupling of aryl aziridines to aryl iodides in an undivided cell, affording a wide range of $\beta$-phenethylamines under mild reaction conditions. The utilization of an electron-rich BiIM ligand is crucial to promoting this transformation. More importantly, using electrons to turn over catalysts showed high efficiency, which provided an alternative method to a combination of TMSCl with a metal reductant system. The role of the ligand structure in improving the reactivity and selectivity of this reaction was elucidated by statistical analysis and was compared to the role of the ligand structure in a mechanistically related photochemical reaction.

## Methods

### General procedure for the electrolysis
In the glovebox, an oven-dried electrochemical cell with a stir bar was charged with aziridine (0.2 mmol, 55 mg, 1 equiv.) and aryl iodide (0.6 mmol, 141 mg, 3 equiv.), NiBr$_2$·glyme (0.02 mmol, 6.2 mg, 10 mol %), ligand **L7** (0.03 mmol, 16.1 mg, 15 mol%), NaI (0.4 mmol, 60 mg, 2 equiv.), triethylamine (0.4 mmol, 2 equiv.), H$_2$O (0.6 mmol, 3 equiv.),

4 mL of DMAc. The tube installed an RVC as the cathode and zinc as the sacrificial anode. The mixture was stirred at 0 °C for 30 min. The reaction mixture was electrolyzed under a constant current of 2 mA at 0 °C until the complete consumption of the starting materials was monitored by TLC (about 18 h). The resulting mixture was diluted with EtOAc and quenched with sat. NH$_4$Cl solution and the aqueous layer extracted with EtOAc (2 × 20 mL). The combined organic layers were dried over MgSO$_4$, the filtrate was concentrated, and the crude product was purified by automated silica gel column chromatography (EtOAc/ hexanes).

More experimental procedures and a photographic guide for Nicatalyzed enantioselective electrochemical reductive cross-couplings are provided in the Supplementary Information.

## Data availability
The X-ray crystallographic coordinates for structures reported in this article have been deposited at the Cambridge Crystallographic Data Centre (CCDC) under deposition number CCDC 2161740 (**3a**). The data can be obtained free of charge from The Cambridge Crystallographic Data Centre [http://www.ccdc.cam.ac.uk/data_request/cif]. The data supporting the findings of this study are available within the article and its Supplementary Information files. The complete datasets used in this paper and the XYZ files are also available at: https://github.com/Milogroup. Data is available from the corresponding authors upon request.

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

## Acknowledgements

This work was financially supported by the National Key R&D Program of China (No. 2021YFA1500100), the NSF of China (Grants 21821002, 21772222, and 91956112), the S&TCSM of Shanghai (Grants 18JC1415600 and 20JC1417100), Syngenta (UK), and Bayer AG (Germany). The authors thank Abigail G. Doyle (University of California, Los Angeles) for her advice.

## Author contributions

Y.-Z.W. and Z.-H.W. designed and performed the experiments. I.L.Z. contributed to the theoretical calculation part. A.M. and T.-S.M. directed the project. B.S., D.L., and Y.-C.G. revised the manuscript. Y.-Z.W., Z.-H.W., A.M., and T.-S.M. wrote the manuscript with input from all authors. All authors analyzed the results and commented on the manuscript.

## Competing interests

The authors declare no competing interests.
