## [Peer Review File · Nature Communications]

REVIEWER COMMENTS

Reviewer #1 (Remarks to the Author):

Mei, Milo and co-workers described an enantioselective nickel-catalyzed electrochemical reductive cross-coupling of aryl aziridines to aryl iodides. A series of structurally diverse chiral amines were prepared in moderate to excellent yields and enantioselectivities. A reasonable reaction mechanism was proposed based on the results of control experiments. However, the same reaction had been reported by Doyle's group (see ref. 47 in this manuscript). The main difference between this work and Doyle's is the reduction system. In this work, electrochemistry was introduced to accelerate the reduction process of Ni, and Doyle et al employed the metal Mn as a reductant. This reviewer thought that this innovation was limited and could not meet the high requirement of Nat. Commun. Furthermore, the following issues should be addressed before the next submission. 1) The reaction reported in ref. 47 should be added in Fig. 1 and typically emphasized in the section of Introduction; 2) the substrate scope should be further studied. Especially, the aliphatic halides and 1,2-disubstituted aziridines should be tested in this reaction; 3) the compound numbers in Fig 4 are not in accordance with those in other sections.

Reviewer #2 (Remarks to the Author):

The manuscript by Mei and co-workers describe an enantioselective nickel-catalyzed electrochemical reductive cross-coupling of aryl aziridines to aryl iodides, affording a wide range of β -phenethylamine under mild reaction conditions. Compared with a combination of TMSCl with a metal reductant system, using electrons to turn over catalysts showed higher efficiency. In the manuscript, the authors elucidate the role of the ligand structure in improving the reactivity and selectivity of this reaction by statistical analysis and compare to the role of the ligand structure in a mechanistically related photochemical reaction, which will have important guiding significance. Thus, I recommend accepting the manuscript after a few minor corrections.

Several minor points for revision are listed below:

1. Page 5, the notes of Fig. 2: "(A) Reaction optimization (B) (D): Voltaic profiles of the (A) Zn/(C) Pt (red) and RVC cathode (black). (C): Cyclic voltammograms recorded on a glassy carbon WE and Pt CE. DMAc containing 0.1 M n-Bu₄NPF₆ and 5mM [Ni] and 5mM Ln. Potentials were calibrated with Fc as an internal standard. (E): Separation of BiIm and BiOx by parameters." should be carefully checked.
2. Page 19, the year in reference 46 is incorrect.

3. Related references on asymmetric electrochemical transformation should be cited: CCS Chem. 2021, 3, 338; J. Am. Chem. Soc. 2022, 144, 6964; Angew. Chem. Int. Ed. 2022, e202210632; Sci. Adv. 2022, 8, eadd7134.

Reviewer #3 (Remarks to the Author):

Mei and co-workers reported an asymmetric electrochemical organonickel-catalyzed reductive cross-coupling of aryl aziridines with aryl iodides. The combination of cyclic voltammetry (CV) analysis of the catalyst reduction potential as well as an electrode potential study provided a convenient route for reaction optimization. And the role of the ligand structure in improving the reactivity and selectivity of this reaction was elucidated by statistical analysis. In addition, asymmetric electrochemical reduction reactions hold great promise in expanding the asymmetric transformations but remains underexplored. However, more detail and evidence should be discussed. With these considerations, though the same reaction was reported by Doyle group before, using nickel/BiOx system in the presence of Mn as the reductant, this work is recommended for publication with Nature Communications after addressing the comments.

1. When using different aryl (pseudo)halides (Supplementary Table 9), the conversion rate were very high but along with very low yield, please comment on this phenomenon and provide the main by-product in the transformations.
2. The optimization of different base and iodides should be more detailed.
3. It seems that the scope of aryl aziridines is limited in this transformation, only five examples were reported, please comment on this result and provide more examples, such as heteromatic aziridines, poly-substituted aziridines, and alkyl aziridines. Please provide few failed examples for the readers to better understand the scope and limitations of the method.
4. Have the authors tried out the alkyl iodides, please provide some examples.
5. Please provide the detailed dimension of the cells used for electrolysis and CV experiments for reproducibility.
6. In Supplementary Fig. 11, the red line and orange line were performed under the same conditions? Why the reductive peaks are different? The reductive peaks reported in Figure 3D of manuscript were not in line with that in Supplementary Fig. 11 (NiBr₂•glyme : L7 : 2a).
7. In Figure 3C, only the reduction peak of NiBr₂•glyme and L7 was observed by the addition of 1a (E_p = -2.71 V) into the mixture of NiBr₂•glyme and L7, and the author draw the conclusion that NiBr₂•glyme and L7 cannot oxidatively add to 1a. Please explain this result, I speculate the overlapped reductive peaks could be observed of 1a and NiBr₂•glyme and L7 if the catalytic process

is involved in this process, in other word, the reductive peak of 1a should be reserved in this process. Besides, a wrong citation on Page 8 in manuscript “These results demonstrate that the catalytic system of NiBr₂•glyme and L7 can oxidatively add to 2a but not 1a.50” Ref. 50 has no relationship with this conclusion.

8. The reported yield and ee of product 3a in Figure 4D (67% and 90% ee) does not match that described in page 9, line 1 (57% and 91% ee). Check carefully again.

9. The structures of product 3 in section 4.1 of SI should be corrected, all of them are racemic.

10. Please check the SI carefully, for example: the title in the manuscript differs from the one in the Supporting Information, and there are two Supplementary Fig. 17 in the SI.

Response to Reviewers

(Manuscript No.: NCOMMS-22-49995)

Reviewer #1:

Question 1: The reaction reported in ref. 47 should be added in Fig. 1 and typically emphasized in the section of Introduction.

Our response: Thanks for your suggestions. The content of Fig. 1 is related to enantioselective electrochemical reductive cross-couplings in our manuscript and the ref. 47 was well introduced and emphasized in the introduction part as shown on page 2.

Question 2: The substrate scope should be further studied. Especially, the aliphatic halides and 1,2-disubstituted aziridines should be tested in this reaction.

Our response: Thanks for your suggestions. Five different aliphatic halides were tested, (2-bromoethyl)benzene could provide the desired product in a 17% yield. No desired product **3** was detected in the case of (bromomethyl)benzene, 2-bromo-2-methylpropane, (chloromethyl)benzene, (2-iodoethyl)benzene, along with the formation of TsNH₂ as the main by-product. When 1,2-disubstituted aziridines such as **1b** and **1c** were employed in this transformation, the ring-opening products were not observed. These results were added to Supplementary Table 13 and Table 11, respectively.

Question 3: The compound numbers in Fig 4 are not in accordance with those in other sections.

Our response: Thanks for your suggestions. The number of aryl aziridine and aryl iodine was reversed, aryl aziridine has been numbered as **1a** and aryl iodine as **2a**. These numbers are uniform in the main text.

Reviewer #2:

Question 1: Page 5, the notes of Fig. 2: " (A) Reaction optimization (B) (D): Voltaic profiles of the (A) Zn/(C) Pt (red) and RVC cathode (black). (C): Cyclic voltammograms recorded on a glassy carbon WE and Pt CE. DMAc containing 0.1 M n-Bu₄NPF₆ and 5mM [Ni] and 5mM Ln. Potentials were calibrated with Fc as an internal standard. (E): Separation of BiIM and BiOx by parameters. " should be carefully checked.

Our response: Thanks for your suggestions. We have checked the note of Fig. 2 and it has been changed as follows: "Reaction optimization. **A** Ligand evaluation. **B** and **C** Voltaic profiles of the Zn/Pt anode (red) and RVC cathode (black). **D** Cyclic

voltammograms were recorded on a glassy carbon WE and Pt CE. DMAc containing 0.1 M *n*-Bu₄NPF₆ and 5 mM [Ni] and 5 mM **L_n**. Potentials were calibrated with Fc as an internal standard. **E** Separation of BiIM and BiOx by parameters.”

Question 2: Page 19, the year in reference 46 is incorrect.

Our response: Thanks for your corrections. We have corrected the published year of reference 46 on Page 19.

Question 3: Related references on asymmetric electrochemical transformation should be cited: CCS Chem. 2021, 3, 338; J. Am. Chem. Soc. 2022, 144, 6964; Angew. Chem. Int. Ed. 2022, e202210632; Sci. Adv. 2022, 8, eadd7134.

Our response: Thanks for your suggestions. These references were cited as References 25-28 in the main text.

Reviewer #3:

Question 1: When using different aryl (pseudo)halides (Supplementary Table 9), the conversion rate were very high but along with very low yield, please comment on this phenomenon and provide the main by-product in the transformations.

Our response: Thanks for your suggestions. The aryl halides bearing electron donating group made they show low reactivity in the oxidative addition to the nickel catalyst. Thus, the starting material was recovered at 32% to 92%, along with the formation of TsNH₂ as the main by-product. These data were added to Supplementary Table 9.

entry	Ar-X	yield (%)	1 (%)	2 (%)	TsNH ₂ (%)
1	Br	16	0	32	25
2	Cl	2	5.1	61	34
3	OTf	0	7.5	80	27
4	OTs	0	10.3	92	35

Question 2: The optimization of different base and iodides should be more detailed.

Our response: Thanks for your suggestions. The detailed optimization of different bases and iodides was added to Supplementary Table 14 and Table 15, respectively.

Screening of Base

Method A

entry	variation from standard conditions	yields (%)	ee (%)
1	DIPEA	83	88
2	Lutidine	27	87
3	DBU	0	0
4	K ₂ CO ₃	39	87
5	Cs ₂ CO ₃	<5	87

Screening of iodides

Method A

entry	variation from standard conditions	yield (%)	ee (%)
1	LiI	78	88
2	KI	42	86
3	ⁿ Bu ₄ NI	59	88

Question 3: It seems that the scope of aryl aziridines is limited in this transformation, only five examples were reported, please comment on this result and provide more examples, such as heteromatic aziridines, poly-substituted aziridines, and alkyl aziridines. Please provide few failed examples for the readers to better understand the scope and limitations of the method.

Our response: Thanks for your suggestions. The heteromatic aziridines, poly-substituted aziridines, and alkyl aziridines were synthesized and examined under the electrochemical conditions, 3-(1-tosylaziridin-2-yl)pyridine could not give any desired product. 2-(naphthalen-2-yl)-1-tosylaziridine, and 2-benzyl-1-tosylaziridine convert to the corresponding ring-opening product in 31% and 28% yield, respectively. We have added the comments for these results in the main text. Moreover, when 1,2-disubstituted aziridines such as **1b** and **1c** were employed in this transformation, the ring-opening products were not observed. These failed examples were added to Supplementary Table 11 and Table 12, respectively.

Question 4: Have the authors tried out the alkyl iodides, please provide some examples.

Our response: Thanks for your suggestions. (2-Iodoethyl)benzene was examined, and no desired product formation. Besides, four different aliphatic halides such as (2-bromoethyl)benzene (bromomethyl)benzene, 2-bromo-2-methylpropane, (chloromethyl)benzene were tested, poor results were obtained along with the formation of TsNH₂ as the main by-product. These results were added to Supplementary Table 13.

Question 5: Please provide the detailed dimension of the cells used for electrolysis and CV experiments for reproducibility.

Our response: The dimension of the cells used for electrolysis and CV experiments was provided in *Photographic Guide for Electrochemical coupling* part in the Supplementary Information.

Question 6: In Supplementary Fig. 11, the red line and orange line were performed under the same conditions? Why the reductive peaks are different? The reductive peaks reported in Figure 3D of manuscript were not in line with that in Supplementary Fig. 11 (NiBr₂·glyme : L7 : 2a).

Our response: Thanks for your questions. Supplementary Fig. 11, shows the interaction between the catalyst and aryl iodine. The red line and orange line were performed under the same conditions, but with different potential ranges. Since the

number of aryl aziridine and aryl iodine were reversed, aryl aziridine should be numbered as **1a** and aryl iodine as **2a**. Thus, the reductive peaks reported in Figure 3D of the manuscript were not in line with that in Supplementary Fig. 11.

Question 7: In Figure 3C, only the reduction peak of NiBr₂•glyme and L7 was observed by the addition of 1a (E_p = -2.71 V) into the mixture of NiBr₂•glyme and L7, and the author draw the conclusion that NiBr₂•glyme and L7 cannot oxidatively add to 1a. Please explain this result, I speculate the overlapped reductive peaks could be observed of 1a and NiBr₂•glyme and L7 if the catalytic process is involved in this process, in other word, the reductive peak of 1a should be reserved in this process. Besides, a wrong citation on Page 8 in manuscript “These results demonstrate that the catalytic system of NiBr₂•glyme and L7 can oxidatively add to 2a but not 1a.50” Ref. 50 has no relationship with this conclusion.

Our response: Thanks for your suggestions. The number of aryl aziridine and aryl iodine was reversed, aryl aziridine has been numbered as **1a** and aryl iodine as **2a**. Therefore, NiBr₂•glyme and L7 cannot oxidatively add to **1a** (aryl aziridine), while they could easily oxidatively add to **2a** (aryl iodine). Moreover, reference 50 was replaced by 51 in the main text.

Question 8: The reported yield and ee of product 3a in Figure 4D (67% and 90% ee) does not match that described in page 9, line 1 (57% and 91% ee). Check carefully again.

Our response: Thanks for your suggestions. The right yield and ee of product **3a** were 67% and 90%, we have corrected the data in the main text.

Question 9: The structures of product 3 in section 4.1 of SI should be corrected, all of them are racemic.

Our response: Thanks for your suggestions. The structures of product **3** in section 4.1 of SI have been corrected into chiral.

Question 10: Please check the SI carefully, for example: the title in the manuscript

differs from the one in the Supporting Information, and there are two Supplementary Fig. 17 in the SI.

Our response: Thanks for your suggestions. The title in the manuscript and Supporting Information were uniform. Besides, the last Supplementary Fig. 17 in the SI has been changed to Supplementary Fig. 20.

REVIEWERS' COMMENTS

Reviewer #1 (Remarks to the Author):

This revised version has good improvement in presentation and well demonstrates the limitations on substrate scopes. Almost all of the issues raised by reviewers were responded to. As my comments gave in the first run of peer review, reporting of the same transformation with literature made the innovation of this work low the average level of Nat. Commun. The results of the substrate scope exploration show that no particular advantages are exhibited in this work when comparing it with that published work. With the consideration of the careful work on ligand study, it is maybe acceptable by Nat. Commun.

Reviewer #3 (Remarks to the Author):

Previous comments have been properly addressed. This work is recommended for publication with Nature Communications.